# Alginate/PVA Polymer Electrolyte Membrane Modified by Hydrophilic Montmorillonite for Structure and Selectivity Enhancement for DMFC Application

**DOI:** 10.3390/polym15122590

**Published:** 2023-06-06

**Authors:** Maryam Taufiq Musa, Norazuwana Shaari, Nor Fatina Raduwan, Siti Kartom Kamarudin, Wai Yin Wong

**Affiliations:** 1Institute of Fuel Cell, Universiti Kebangsaan Malaysia, Bangi 43600, Malaysia; p105749@siswa.ukm.edu.my (M.T.M.); p112062@siswa.ukm.edu.my (N.F.R.); ctie@ukm.edu.my (S.K.K.); waiyin.wong@ukm.edu.my (W.Y.W.); 2Faculty of Engineering & Built Environment, Universiti Kebangsaan Malaysia, Bangi 43600, Malaysia

**Keywords:** alginate, PVA, montmorillonite, membrane, DMFC

## Abstract

Nafion is a commercial membrane that is widely used in direct methanol fuel cells (DMFC) but has critical constraints such as being expensive and having high methanol crossover. Efforts to find alternative membranes are actively being carried out, including in this study, which looks at producing a Sodium Alginate/Poly (Vinyl Alcohol) (SA/PVA) blended membrane with modification by montmorillonite (MMT) as an inorganic filler. The content of MMT in SA/PVA-based membranes varied in the range of 2.0–20 wt% according to the solvent casting method implemented. The presence of MMT was seen to be most optimal at a content of 10 wt%, achieving the highest proton conductivity and the lowest methanol uptake of 9.38 mScm^−1^ and 89.28% at ambient temperature, respectively. The good thermal stability, optimum water absorption, and low methanol uptake of the SA/PVA-MMT membrane were achieved with the presence of MMT due to the strong electrostatic attraction between H^+^, H_3_O^+^, and ^−^OH ions of the sodium alginate and PVA polymer matrices. The homogeneous dispersion of MMT at 10 wt% and the hydrophilic properties possessed by MMT contribute to an efficient proton transport channel in SA/PVA-MMT membranes. The increase in MMT content makes the membrane more hydrophilic. This shows that the loading of 10 wt% MMT is very helpful from the point of view of sufficient water intake to activate proton transfer. Thus, the membrane produced in this study has great potential as an alternative membrane with a much cheaper cost and competent future performance.

## 1. Introduction

During this decade, it is anticipated that fuel cell systems will be able to produce energy through electrochemical reactions without involving the combustion process. Depending on the fuel utilized, different types of fuel cells have been developed, and one system that has attracted a lot of attention is the direct methanol fuel cell (DMFC) [1]. DMFC produces maximum power density at room temperature using methanol as fuel, with an efficiency of up to 60%. The fact that the fuel used in DMFC is liquid rather than hydrogen gas, which is more difficult to carry and store, is a significant benefit of DMFC over PEMFC [2]. There are several applications for DMFC, including portable and mobile ones. The exceptional potential of this DMFC is still not fully realized, however, as its commercialization is moving very slowly because of some serious problems. The main problems include extremely high production costs, which are related to the expensive component [3].

The commercial membrane Nafion is utilized in DMFC, although it is highly expensive, costing up to $1000 per square foot [4]. Nafion is made of non-biodegradable-based materials and has other drawbacks such as high methanol permeability, which results in fuel loss in the DMFC system. Despite Nafion’s outstanding proton conductivity and great chemical stability, the DMFC system faces significant challenges because of the difficulties mentioned above [5]. Due to its significant function in the DMFC system, the polymer electrolyte membrane (PEM) is an extremely significant component. PEM serves as a separator between fuel and oxidant at the anode and cathode sections, respectively. It also serves as the primary route for protons while simultaneously blocking the passage of electrons. The DMFC system’s efficiency will be strongly influenced by membrane efficiency [6,7].

Excellent proton conductivity, low methanol crossover, high mechanical and chemical stability, and affordability are the essential characteristics that an ideal membrane must have [8]. Alginate (natural) and poly (vinyl alcohol) (PVA) (synthetic) have shown significant potential in the creation of blends or composites for a variety of applications. Biopolymer-based membranes are gaining attention since they are natural, affordable, and prevalent in the environment, and efforts to find membrane substitutes are in full swing [9]. Alginate has been investigated in several areas, including food packaging, biomedicine, tissue engineering, and medication delivery [10,11]. The two biggest units in alginate, a brown seaweed species that is a member of the family of water-soluble polysaccharides, are 1–4)-linked D-mannuronic acid (M) and L-guluronic acid (G). It is highly possible that it can absorb 200–300 times more water than it weighs in total. Thus, the super hydrophilic properties that belong to alginate degrade the mechanical properties of the alginate membrane [12]. Alginate has good potential, but it contains far fewer transfer pathways for protons and very low proton conductivity. Studies from the past suggest a variety of strategies for the future to enhance alginate’s capabilities and overcome its drawbacks, including combining it with other polymers and adding fillers to its matrix. In comparison to pure alginate, composite materials have turned out to have better qualities in terms of thermal and mechanical stability, as well as proton conductivity [13].

Due to its low cost, hydrophilicity, suitable mechanical properties, and biocompatibility, PVA has been widely used in several industry sectors [14]. It also exhibits outstanding chemical rigidity and physicochemical features. According to earlier research, alginate is thought to be compatible with a number of substances that have higher proton conductivity values, such as chitosan–sodium alginate (4.2 × 10^−2^ S cm^−1^) [15], alginate–carrageenan (3.16 × 10^−2^ S cm^−1^) [16], PVA–sodium alginate (9.1 × 10^−2^ S cm^−1^), alginate–GO (13.2 × 10^−3^ Scm^−1^) [17], alginate–AlO_3_ (25.6 × 10^−3^ S cm^−1^) [18], and alginate–TiO_2_ (17.3 × 10^−3^ S cm^−1^) [19].

MMT is a common inorganic filler that has been widely utilized as an additive in Nafion membranes for fuel cell applications. Due to its layered structure and high aspect ratio, MMT dramatically decreased methanol permeability when it was incorporated into Nafion membranes. A barrier to the methanol route has developed efficiently with organized layers of MMT [20]. According to Kim et al. [21], 10 wt.% of MMT in Nafion led to the lowest methanol permeability and the highest proton conductivity. With the creation of the amorphous phase and a decrease in the glass transition temperature (T_g_), the hydrophilic MMT has been a significant factor in the rise in proton conductivity and the selectivity of the membrane. At the same time, MMT has strong stiffness characteristics that can regulate the interaction and lower the membrane swelling ratio to maintain the mechanical strength of the membrane [22].

Due to the great potential of MMT, this study investigates the influence of MMT organic filler in the alginate/PVA copolymer blended membrane. The membrane thin film was produced by using a simple blending and solution casting method. Based on our knowledge, this is the first time that an SA/PVA-MMT composite membrane has been developed and used for DMFC applications.

## 2. Chemical and Methods

White solid sodium alginate fine powder and white solid MMT granules were both obtained from Acros Organics manufacturer, in Greci, Belgium originally. White solid Polyvinyl alcohol coarse powder and white solid calcium chloride pellets were purchased from R&M Chemicals in Selangor, Malaysia. These chemical powders were ready-for-analysis grade. Methanol (CH_3_OH, 99.7%) was obtained from Merck, manufactured in Darmstadt, Germany, and glutaraldehyde (OCH(CH_2_)_3_CHO, 25%) was purchased from Sigma Aldrich, manufactured in Darmstadt, Germany as well. These chemicals were used directly without any further purification. Throughout the experiment, deionized water was used from a Millipore system (Milli-Q) of Thermo Scientific (Smart2Pure 3 UV Barnstead Smart2Pure Water Purification system), Selangor, Malaysia.

### 2.1. Synthesis of SA/PVA Biopolymer

The SA/PVA polymer matrix was first synthesized by preparing PVA and SA solutions separately. The ratio of SA:PVA used is 40:60, according to the best ratio of Yang, Wang, and Chiu (2014) [23], whereby PVA is ensured not to leave any undissolved solids on the bottom of the beaker surface by setting the condition of deionized water to 90–100 °C to dissolve PVA, and the temperature of deionized water to 70 °C to dissolve SA. The dissolution period of both SA and PVA polymers took 3–5 h. The hot PVA solution is then mixed into the SA solution, before being cross-linked with 1 mL of 10% GA solution. The solution was stirred vigorously for 30 min to ensure that the mixture was homogeneous. This mixture is then left to even out in the sonicator for 30–60 min, also aiming to remove foam.

### 2.2. Synthesis of SA/PVA-MMT Blended Membrane

The solution of the MMT fillers was added (various loading of 2–20%) into a beaker of the GA-internal crosslinked-polymer solution (SA/PVA) and stirred overnight, before being sonicated for 6 h at room temperature. After being thoroughly mixed, this 30mL of SA/PVA-MMT solution was carefully poured into a petri dish of 90mm diameter. To note, the poured volumes were fixed to maintain the membrane thickness at around 144 μm. It is necessary to ensure that there are no bubbles on the surface of this solution so as to not affect the characterization analysis and performance tests later. This sample solution is then dried in a furnace at a temperature of 60 °C for 12 h, followed by annealing at 80 °C for 1 h to ensure that the membrane is completely dry without making it an inflatable structure but maintaining its flexible properties. The final appearance of the membrane is in the form of a thin film.

### 2.3. SA/PVA-MMT External Cross-Linking

The dried SA/PVA-MMT membrane film was then subjected to external cross-linking by immersing the membrane in a mixed solution of 1.5 wt% calcium chloride and 10% glutaraldehyde for 60 min (to act as a binder for SA/PVA chain), before being tested with a membrane performance test. Uncross-linked membranes were sent for characterization analysis. This external cross-linking is very important to prevent the membrane sample from dissolving in water for further membrane performance tests. The dissociated calcium ions would react with the oxygenated group of GA to become a perfect combination of crosslinking agents in binding together with the SA/PVA chain so that it remained insoluble in water. After the one-hour immersion, the membrane surface was washed with deionized water to remove any cross-linking agent that was no longer bound and left to dry under a fume chamber for about 4 h. This membrane surface was ensured to remain flat during the drying period. Table 1 shows the membranes and their material contents.

### 2.4. Membrane Characterization

The functional groups contained in pure alginate and filled alginate-based membranes were analyzed by FTIR (PERKIN ELMER), the wavelength range of which is in the range of 550–4000 cm^−1^, and Raman spectroscopy for metal-containing membrane samples. The morphology of the film and the internal structure of the membrane were determined through a field-emission scanning electron microscope (FESEM, ZEISS SUPRA 55VP), with a 5 kV operating voltage. This FESEM analysis is also combined with energy dispersive spectrophotometry (EDX, FEI QUANTA 400 FESEM), which aims to identify the percentage of each element of the membrane composite. The thermal and chemical stability of the membrane was analyzed by a thermal stability instrument (STA 6000 TGA, Waltham, MA, USA). The microstructure of the composite membrane was determined through XRD analysis. This diffraction peak pattern was obtained using the D8 Advance XRD model, Bruker AXS Germany. The amorphous nature of the membrane is aided by contact angle analysis, from which the hydrophilic and hydrophobic properties of the membrane can be traced to specific materials.

### 2.5. Membrane Performance Tests

#### 2.5.1. Fluid Intake and Swelling Ratio

The rate of water absorption and swelling ratio by the membrane is an important yardstick in this study, which can be determined by measuring changes in the weight and thickness of the membrane in dry and wet conditions. The membrane was soaked in deionized water for 24 h at room temperature, then the wet weight and membrane thickness readings, *W_wet_* and *t_wet_*, were recorded. The weight and thickness readings of the dry membrane, *W_dry_* and *t_dry_*, were taken after 24 h of drying at room temperature.
(1)$$WU\%=\frac{W_{w}-W_{d}}{W_{d}}\times 100$$
(2)$$SW\%=\frac{t_{w}-t_{d}}{t_{d}}\times 100$$

To determine the water uptake (*WU*%) and swelling ratio (*SW*%) of the membrane, Equations (1) and (2) are used. The calculation of methanol intake is the same as Equation (1) and the method is the same as water intake, only the deionized water immersion solution is changed to a 10% methanol solution.

#### 2.5.2. Proton Conductivity

Membrane proton conductivity was calculated based on the formula of the membrane resistance value obtained through a potentiostat machine. The membrane needs to be soaked in deionized water to activate protons and any ions for 24 h before starting the test. The wet membrane helps the proton flow process. The formula for estimating proton conductivity is as follows (3):(3)$$\sigma =\frac{L}{RWT}$$
where *L*, is the length of the sample (cm), *R* is the resistance value obtained from the Potentiostat machine, *T* is the thickness of the membrane using a micrometer screw gauge (mm), and *W* is the width of the membrane sample tested (cm) [24].

#### 2.5.3. Methanol Permeability

The methanol permeability test of this membrane aims to test the extent to which the potential of this membrane can prevent the cross-transfer of fuel by using two parts of the liquid tank; the membrane is placed midway between these two tanks, where the tank labeled A is filled with 135 mL of 2 M methanol while the tank labeled B is filled with 135 mL of deionized water. This performance test was carried out for 36 h with the rotation of a magnetic bar of uniform force at room temperature; the methanol will slowly move across the membrane depending on its permeability characteristics. Thus, a good membrane will show low methanol permeability. The permeability of methanol to tank B can be measured through the refractive index using a refractometer. The refractive index readings are read at 5 h intervals in order to observe the changes. All the refractive index readings were recorded and converted to concentration units (mol/L) through a standard graph of refractive index versus methanol concentration. The value of methanol concentration at a certain moment is then used in Formula (4) to obtain methanol permeability:(4)$$P=\frac{1}{C_{a}}\times \frac{\Delta C_{b}(t)}{\Delta t}\times \frac{LV_{b}}{A}$$
where the symbol *Ca* indicates the concentration of methanol solution (mol/L), Δ*C_b_(t)* is the concentration of deionized water that changes with time (mol/L), Δ*t* is the last time the refractive index reading was taken (s), *L* is the thickness of the membrane, *V_b_* indicates the volume of the solution in tank *B* (L), and A indicates the area of the tank hole that touches the membrane surface (cm^2^).

#### 2.5.4. Membrane Selectivity

Membrane selectivity can be determined by the ratio of proton conductivity and methanol permeability. The high selectivity value indicates that the membrane has a high potential to be used in the DMFC system. The selection value is calculated with the following formula:(5)$$\varphi =\frac{\sigma }{P}$$
where *φ* indicates selectivity, *σ* indicates proton conductivity and *P* indicates methanol permeability.

## 3. Results and Discussion

### 3.1. Fourier-Transform Infrared Spectroscopy (FTIR)

Figure 1 shows the FTIR spectrum of the SA/PVA sample, SPM2-20. There are several significant peaks that indicate the presence of functional groups which has influenced the membrane performance test. At the broad 3000–3500 cm^−1^ peak, it shows the stretching of the O-H hydroxyl group present in all samples, which means that all these membranes can allow the flow of water molecules and activate proton conductivity. This spectrum also proves that there is a similar peak at 2700–3000 cm^−1^, which indicates the occurrence of stretching on the C-H group, meaning that all membranes are based on hydrocarbons [24,25]. In the fingerprint part, it is clearly seen that there is a change in the peak intensity starting at 1400–1700 cm^−1^ and 700–1200 cm^−1^ which, respectively, represent COO^−^ stretching and C-C, C-O, and C-H bending. Table 2 lists the significant peaks along with their wave numbers for each membrane sample [24].

On the stretch vibration side, the peaks for each sample did not show significant intensity changes. The peak of the spectrum in the bending vibration part experiences a clear intensity change when the clay filler is inserted into the SA/PVA membrane. A weak peak at wave number 1739 cm^−1^ appears in the SA/PVA sample, indicating the C=O stretching contained in the polymer compound [26]. This peak gets smaller when the MMT filler is added, indicating that the interaction has occurred between the biopolymer matrix and the inorganic filler phase, then the C=O stretch becomes weaker and harder to detect by infrared compared to other influential functional groups [27]. At 1085 and 824 cm^−1^ peaks, moderate intensity peaks representing C-O stretching of alcohol remained the same in SA/PVA, SPM2, and SPM5. However, the intensity changed after it occurred in SPM10, SPM15, and SPM20. The same is true of the 917 and 843 cm^−1^ peaks, which represent silanol, Si-O, and aluminol Al-O bending in SPM10, SPM15, and SPM20 [28]. This peak is not significant at SPM2 and SPM5 because the MMT load does not have a strong bending. The importance of these silanol and aluminol groups relates to membrane performance; the presence of these groups defines MMT as an inorganic filler able to reduce methanol entry and membrane swelling.

The change in the intensity of these peaks clearly shows the strength of the stretching and bending of functional groups contained in this composite membrane. The SA/PVA polymer combination clearly contains oxygenated functional groups such as O-H, C-O, C=O, C-O-C, and -COO. This combination of SA/PVA further broadens the hydroxyl peak, meaning that the stretching vibration of the molecules in SA/PVA is weaker than that of pure SA powder; this condition facilitates the formation of hydrogen bonds between the O-H groups of SA and PVA [29]. The oxygen-carrying particles contained in -COO have a reactive double π bond, which will provide exceptional results for conducting protons [17]. The change in the wave number shift of the -COO group (from 1599, 1414 cm^−1^ to 1657, 1418 cm^−1^) and the reduction of the peak intensity also indicate the occurrence of a reaction with the MMT reactive group [30]. The reduction in the intensity of the 1739 cm^−1^ SA/PVA peak at the peaks of SPM2-20 means that the C=O stretching vibration interacts with the MMT element and forms a stronger and more stable bond [31].

The presence of MMT has more impact with the presence of Si-O and Al-O functional groups, which appear at 918 and 843 cm^−1^ and are more clearly seen in the spectra of SPM10, SPM15, and SPM20. The intensity of this peak is very weak, and there is a slight shift when compared to the study of [29] (1000–1300 cm^−1^) and [31] (1048, 470 cm^−1^), due to its affinity to interact with SA/PVA combined polysaccharides. The interaction that occurs between the polymer and the MMT filler even strengthens the C-O-C and O-H bending on each tetrahedron and octahedron layer. This oxygenated group helps increase proton conductivity [32], with this MMT clay element being able to reduce methanol permeability [33], making this composite membrane have a high potential to produce good PEM performance.

**Table 2 polymers-15-02590-t002:** Infrared peaks identified by wavenumbers.

Functional Groups	Wavenumbers (cm^−1^)						References
	SA/PVA	SPM2	SPM5	SPM10	SPM15	SPM20	
Vibration O-H SA/PVA	3260	3259	3257	3264	3262	3265	[19]
Vibration C-H SA/PVA	2939	2938	2938	2938	2939	2938	[29]
C=O	1739						
-COO asymmetrical & symmetrical SA/PVA	1599, 1414	1600, 1413	1600, 1413	1598, 1412	1657, 1418	1657, 1418	[19]
C-O alcohol/ether	1230	1328	1327	1325	1330, 1234	1329, 1234	
Polysaccharide bending C-C, C-H	1085, 828	1086, 825	1086, 827	1085	1085	1085	[29]
C-O-C	1032	1031	1032	1029	-	-	[25]
Si-O, Al-O MMT	-	-	-	947, 822	918, 843	917, 843	[31]

### 3.2. Raman Spectroscopy

Raman spectroscopy emits vibrations that modulate the polarizability of a sample object, especially for hydrocarbon samples containing metallic materials. Figure 2 shows the Raman spectra of SPM2, SPM10, and SPM20 membranes for this study. Through these three membranes, the difference in the peak intensity of each membrane can be seen significantly, especially at the shift of 1440–1600 cm^−1^, which represents C-H and O-H bending, as well as C-C stretching on PVA-MMT [34], while the peak equation can clearly be seen to occur at the displacement of 850–910 cm^−1^, which is C-C stretching. In this degree of crystallinity of PVA can also be understood, and is at a displacement of 1145 cm^−1^ according to [34]. In general, the difference in peak the intensity that occurs indicates the way the presence of MMT is increasing when the peak intensity also increases. In conclusion, based on this Raman analysis, the SPM composite membrane has great potential to become a stronger membrane with the presence of semi-metals in the MMT clay filler and the semi-crystalline nature of PVA, greatly improving the weakness of the SA biopolymer.

### 3.3. X-ray Powder Diffraction (XRD)

Figure 3 above illustrates the peak trend of the diffractogram according to the concentration ratio of the MMT fillers. It can be observed that the semi-crystalline nature of PVA was successfully proven by the appearance of a high peak at an angle of 19–20° [35,36] and a small amorphous peak at an angle of 2θ 39–40°, which is the peak of similarity in all samples of this hybrid membrane. According to [37], there is an MMT crystal peak at an angle of 2θ = 5.82° with a silica layer spaced 1.51 nm apart in the d001 crystal plane. The MMT crystal peak in Figure 3 appeared at a higher angle, which is 8.76°, and is clearly visible in the SPM2 sample. The appearance of this peak is similar to the studies performed by [38,39], who obtained 2θ = 8.88° and 8.56°, respectively, with a d_001_ distance = 10.43 Å. From the observation, the decrease in the intensity of the MMT crystal peak area, which occurred at SPM5, SPM10, SPM15, and SPM20, shows that the MMT crystal phase has changed to amorphous. Similarly, the angle 2θ = 5.56°, which should be the peak of pure SA crystals, has decreased in intensity when mixed with PVA polymer [39]. SPM2 displays a small amorphous peak of PVA at a lower diffraction angle of 26.71°, then decreases as the MMT loading increases. This is due to the hindrance effect of the clay when forming a composite structure with the biopolymer. The shift of this peak to a lower angle indicates that the distance between the layers has grown as the polymer chain successfully intercalates into the polymer structure [40], whereas the remaining range without significant peaks indicates the amorphous part found in this hybrid membrane sample. Influential substances in this membrane that contribute to the amorphous nature are believed to be sodium alginate and MMT, the contribution of which can be attributed to their hydrophilic properties that are able to conduct protons. Through this XRD, the metals contained in the membrane can be known, such as sodium, aluminum, silica, calcium, potassium, and magnesium. This XRD analysis clearly proves that the SPM hybrid membrane tends to be amorphous and very suitable for conducting protons.

### 3.4. Field Emission Scanning Electron Microscopy (FESEM)

For the micrograph image, only four samples were analyzed because the morphological analysis needs to look at the change in trend when the clay filler load increases, namely in the SA/PVA, SPM5, SPM10, and SPM20 samples. These membranes have been proven to have elements of carbon, oxygen, and sodium, as can be seen in EDX in Figure 4C,F,I,L. Figure 4A,D,G,J shows the morphology of the membrane surface, while Figure 4B,E,H,K depicts the cross-section of the membrane.

Based on Figure 4A, which represents SA/PVA, the surface image appears to have pores, as can be seen through the cross-sectional image (Figure 4B). Excessive pores are not suitable because they create pathways for water absorption which will result in excessive swelling of the membrane [8]. When adding a small amount of MMT, the surface of SPM5 appears rougher than SA/PVA, as illustrated in Figure 4D. The cross-sectional image of SPM5 also appears coarser, indicating the successful intercalation of MMT into the polymer matrix, as evidenced by the EDX elemental peaks, guided by the increased peaks of Mg, Al, Si, and K. However, the decreased peaks of Na are due to the mixing of MMT with the polymer chain, disrupting the Na-containing polymer structure [24]

As the MMT increases, the surface of the SPM10 membrane appears rougher and its cross-sectional area is denser, as shown in Figure 4G,H. This indicates that the water molecules have completely evaporated from the polysaccharide matrix [25] because the fine white particles seen in Figure 4H show the successful intercalation of MMT into the SA/PVA biopolymer matrix. However, the resulting agglomeration in SPM20 is due to the addition of excessive MMT which, in turn, results in it being concentrated in one place. Therefore, loading below 20 wt% MMT is more suitable to be incorporated in the composite membrane because of the homogenous textured surface produced, and this is further discussed in the liquid uptake section.

Finally, upon adding a matching amount of MMT, both the viscous hydrophilic SA/PVA polymer and the fine-particle hydrophilic MMT were completely mixed and compatible without any phase separation. The appropriate amount of MMT would facilitate the reduction of the passage of methanol molecules and thus increase the ion conductivity in the composite membrane.

### 3.5. *Thermogravimetric Analysis* and Differential Scanning Calorimetry (TGA-DSC)

Referring to Figure 5a, the polymer mixture begins to degrade at 220–240 °C, proving that it is better than the previous study, which showed that the first stage starts at 80 °C [23] and 198 °C [18] for the SA and PVA mixture only. The occurrence of this first degradation phase is due to solvent evaporation [34]. By adding MMT, the membrane has a combination of amorphous and crystalline phases, making the degradation phase slightly slower. This polymer must have a semi-crystalline phase because high amorphous properties can result in low film structural strength. The presence of semi-crystalline properties of PVA can further strengthen the hydrogen bonding interaction between the polymer chain and the filler bond. The addition of the crosslinking agent, CaCl_2_, and the GA solution also improve its thermal stability, as well as the annealing process during the membrane film drying. The weight loss value of the membrane is around 50.4–55.2%, which starts in the range of 259.8–263.5 °C, which is dominated by SPM15. The second degradation phase for this MMT-filled membrane has a lower thermal stability compared to SA/PVA at a temperature of 307 °C [23]. This is because of the presence of MMT, which is hydrophilic, which makes it easy to degrade. The hydrophilic nature allows the membrane to retain its water molecules in the matrix, then degradation will occur at a lower temperature [17,18]. Nevertheless, the presence of PVA and MMT in the sodium alginate-based membrane was much better, which is because this blended membrane undergoes a third degradation phase that has a higher range (470–800 °C) to break the polysaccharide backbone polymer chain, that contained the interfacial interaction with improved crystallinity.

The DSC thermogram in Figure 5b above shows that the glass transition phase changes as the MMT loading increases. The glass transition temperature increases from SA/PVA to SPM10, which is due to the reduced dipole interaction that has occurred between the homopolymers [41]. This coincides with the phase of the TGA thermogram during which the hybrid membrane faces a phase change to a rubbery state at this temperature when the clay filler is contained. However, at a concentration of 15 wt% MMT in this hybrid membrane, the glass transition temperature decreased to a complex peak value of 123.6 °C, where it can be observed that the peak collides with the control membrane, SA/PVA, in the thermogram. This indicates that the polymer backbone softens to a much higher extent. In relation to water uptake performance, SPM15 absorbed more water molecules than SPM10, up to an increase of 10.84%, indicating that it contains voids to bind water. The presence of the void makes the SPM15 membrane more easily degraded and phase changed. Therefore, SPM10 is the most favorable membrane in this study because it obtains thermal stability with the best glass transition temperature value, 196.7 °C.

### 3.6. Contact Angle

Figure 6 depicts a drop of water on a membrane sample to determine the value of the contact angle. Figure 6b and 6c shows the presence of increased MMT, making the membrane more hydrophilic with a reduction in the contact angle value to 49.22° at a loading of 10 wt% MMT, from 60.58° at a loading of 5 wt% MMT. Meanwhile, the increase in the contact angle from 56.81° indicates that the inclusion of this MMT clay reduces the hydrophilic properties of the membrane because the larger particle size of this clay filler covers some of the passage space of water molecules. However, the increase in MMT content makes the membrane more hydrophilic. This concludes that the loading of 10 wt% MMT is very helpful from the point of view of sufficient water intake to activate proton transfer.

### 3.7. Performance Test

#### 3.7.1. Physical Insights

Figure 7 shows the physical observations of each synthesized membrane sample. These six membranes were cut according to certain size measurements before starting the membrane performance test. The performance test requires a 2 × 2 cm sample for water uptake and swelling ratio, while a 4 × 1 cm sample for proton conductivity and a 3 × 3 cm sample for methanol permeability are required. All these membranes have a thickness of around 144 ± 35 μm, which was controlled by the poured solution volume into the petri dish being fixed to 30 mL. The observation on the physical appearance of the SA/PVA membrane containing this MMT clay looks yellowish white and harder as the MMT increases.

#### 3.7.2. Fluid Intake and Swelling Ratio

For water uptake performance, based on Figure 8, the membrane starts to absorb less water at 2 wt% MMT. Then, the trend started to decline again at 15 wt% MMT after a slight increase at 10 wt% MMT. The increase in water absorption is due to the inclusion of MMT into the pure SA/PVA membrane, and slightly due to the hydrophilic nature of MMT, which provides a high-water content to the membrane.

This hydrophilic nature is due to the oxygen functional group (-OH) [31] due to which the abundant water creates a continuous transfer channel and facilitates the movement of ions [24]. These functional groups interact with each other (polymer and filler) through strong hydrogen bonds or polar–polar interactions [31]. The lowest water intake point is at 20 wt% (SPM20) after 10 wt% (SPM10) MMT. The level of water intake decreased from 162.46% to a value of 129.65% at 10 wt% before it decreased again to 74.89% in SPM20.

At the highest MMT content, agglomeration occurred, which has been proven in the FESEM analysis in Figure 8. As a result, ion channels are reduced and there is a restriction of water absorption. Figure 8d,e,g,h,j,k illustrates a reasonable situation of how MMT fills the polymer matrix. The filler in the membrane has dispersed homogeneously so that it can reduce the free voids in the matrix [31]. SA/PVA polymer interacts with MMT particles according to its ionic or non-ionic character. Ionic polymers induce electrostatic interactions, while non-ionic polymers are adsorbed on clay mineral surfaces through steric interactions [29]. The SA/PVA movement chain then decreases, leading to a decrease in water content. As well as this, the decrease in water content resulting in the swelling ratio of the membrane decreasing would benefit its mechanical properties [42]. Based on the experimental results as shown in Table 3 and Figure 8, the value of methanol uptake decreased after reaching its peak at 2 wt.% MMT, from 203.3% to 53% at 20 wt.% MMT. The explanation for methanol uptake is similar, as the graph trend decreases as MMT loading increases. The same trend shows that there is a network and a good bond between the alginate polymer and MMT, which prevents the entry of excess fuel [24].

#### 3.7.3. Proton Conductivity and Methanol Permeability

Proton conductivity in a membrane is closely related to water intake because the activation of water molecules will form positively charged ions (protons) and negatively charged ions (electrons). Therefore, the water path in the membrane is also a proton path. Figure 9 combines the results of proton conductivity and methanol permeability to easily compare the performance of the membranes, which is suitable to answer the problems of this study.

Figure 9 shows a non-parallel graph trend between proton conductivity and methanol permeability. The proton conductivity value fluctuates when the MMT load increases in the membrane. The maximum value of proton conductivity successfully achieved is 8.0510 mS/cm, produced by SPM20. When conducting a conductivity test, the important functional groups involved are H^+^, H_3_O^+^, and ^−^OH ions, of which, H^+^ is dissociated from water molecules, jumping from one molecule to another. The lack of a single electron causes H^+^ to actively react with free ^−^OH ions of MMT, which also helps the proton shift along this medium. The hydration state of the membrane is very important because it can produce more H^+^ that will actively bind with the lone pair of electrons in H_2_O, then form a highly reactive ion, H_3_O^+^. This cycle of proton transport through negatively charged ions repeats simultaneously with proton hopping. In this study, it is believed that both proton conduction mechanisms occur, that is, the Grotthuss mechanism (protons jump) and the Vehicular mechanism (protons are transported) [17]. However, the tendency is towards the Grotthuss mechanism due to the presence of hydrogen bonds and acid—base pairs that help the proton to jump. The mechanism of Grotthuss-type proton conductivity, as shown in Figure 10. Water absorption also decreases when the MMT increases (if guided by the water intake findings in the previous section), and then it does not favor the Vehicular mechanism [17].

The proton conductivity decreases to 2.03 mS/cm, as produced by SPM15, after reaching a peak of 6.50 mS/cm by the previous SPM10. The phenomenon that occurs is the reduction of the number of free volumes in the membrane because it is increasingly filled by MMT, because of which it is difficult for protons to be transferred. Nevertheless, the tendency of proton conductivity to rise at 20 wt% MMT is due to the presence of active sites on MMT increasing (referring to the intensity peak of the FTIR spectrum) and molecular distribution density which, in fact, also helps proton jump [43]. These proton conductivity values are lower for SPM5 and SPM15, which is caused by the activation of the blocking properties of MMT when the overload will have an aggregate effect on the membrane [17].

On the other hand, the methanol permeability of the SPM membrane dropped after 2wt% MMT, at the lowest value of 1.27 × 10^−8^ cm^2^/s, as shown by SPM10, and rising back after 10wt% MMT. The decrement values were due to the fulfillment of free voids by MMT particles which activate the blockage properties of MMT. Meanwhile, the increment values were owing to its hydrophilicity properties that enable the methanol permeation as well as proton diffusion [43].

### 3.8. Membrane Selectivity

Based on Figure 11, the selectivity of the SA/PVA-MMT hybrid membrane is the highest at the concentration of 20 wt% MMT (6.13 × 10^5^ S s/cm^3^), followed by 10 wt% MMT (5.11 × 10^5^ S s/cm^3^). The values for water uptake, methanol, and swelling ratio were also the lowest for 20 wt% MMT compared to others. However, based on the FESEM analysis in Figure 4, the surface morphology of the hybrid membrane of 20 wt% MMT appears to be excessively agglomerated compared to 10 wt% MMT. A large amount of MMT helps to close the pores in the membrane so that the permeability and methanol uptake can be reduced and allow the passage of protons only. However, an excessive amount of MMT also poses a high risk to the membrane surface, resulting in the formation of a single-phase separated phase, in which the distribution of fillers and polymer chains is uneven throughout the membrane, causing steric effects on the interaction between molecules and reduced effectiveness on DMFC performance. Table 4 listed the results for proton conductivity, methanol permeability and selectivity for each membranes studied.

## 4. Conclusions

As a final summary, this study succeeds in preparing an alternative membrane via a simple casting method which consists of alginate and PVA as copolymers and MMT as the nanofiller. The optimum membrane has shown excellent proton conductivity and low methanol permeability, which consequently contribute to significant selectivity. The presence of MMT augmented the major properties of the alginate-based polymer membrane in terms of proton conductivity and methanol permeability. The maximum value of proton conductivity that was successfully achieved is 8.0510 mScm^−1^, produced by SPM20. When conducting a conductivity test, the important functional groups involved are H^+^, H_3_O^+,^ and -OH ions, of which H^+^ is dissociated from water molecules, jumping from one molecule to another. In addition, the presence of MMT provided a barrier effect, thus reducing the methanol crossover through the membrane. The selectivity of the SA/PVA-MMT hybrid membrane is the highest at the concentration of 20 wt.% MMT (6.13 × 10^5^ S scm^−3^). This value should be a good benchmark for alternative membrane performance, and it is comparable with other membranes, especially biopolymer-based materials.

## Figures and Tables

**Figure 1 polymers-15-02590-f001:**
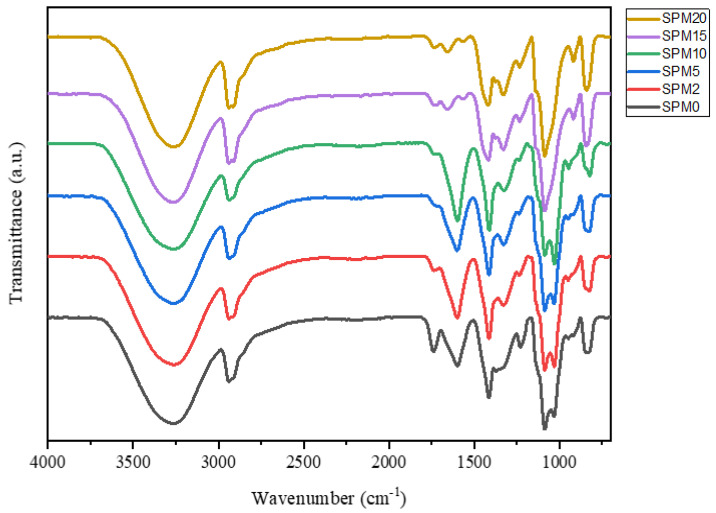
FTIR analysis for SA/PVA-MMT membrane with various loading of MMT.

**Figure 2 polymers-15-02590-f002:**
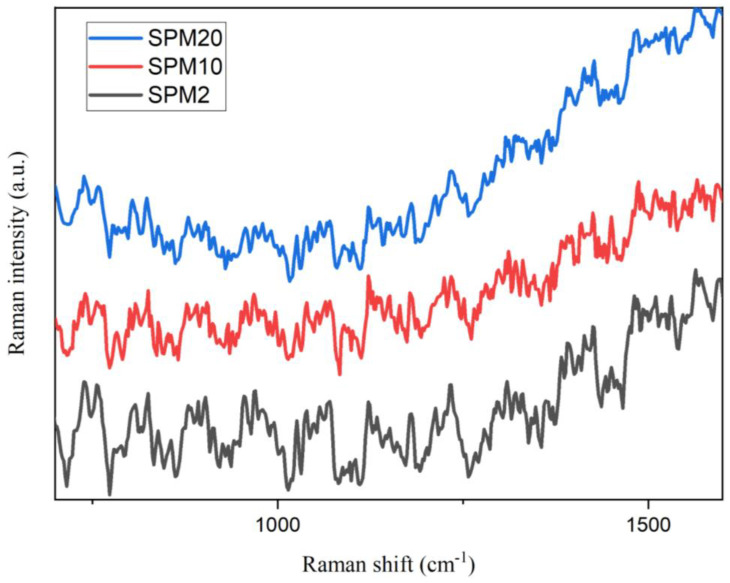
RAMAN analysis for membranes SPM2, SPM10, and SPM 20.

**Figure 3 polymers-15-02590-f003:**
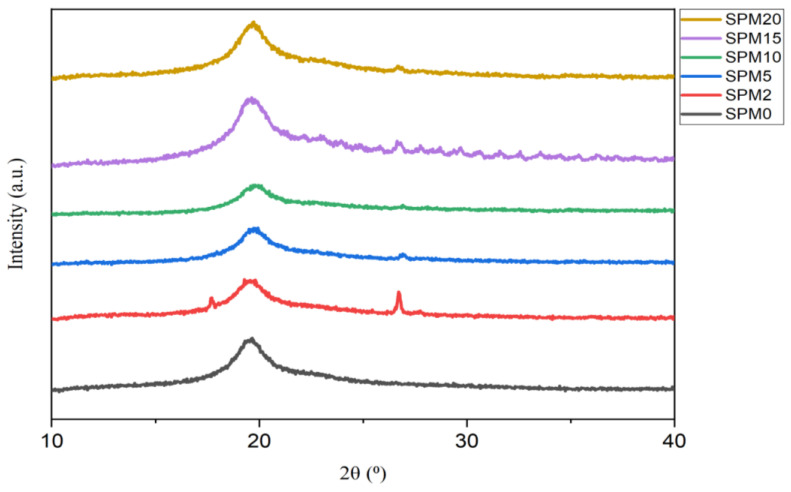
XRD Analysis for SA/PVA-MMT blended membrane with various loading of MMT.

**Figure 4 polymers-15-02590-f004:**
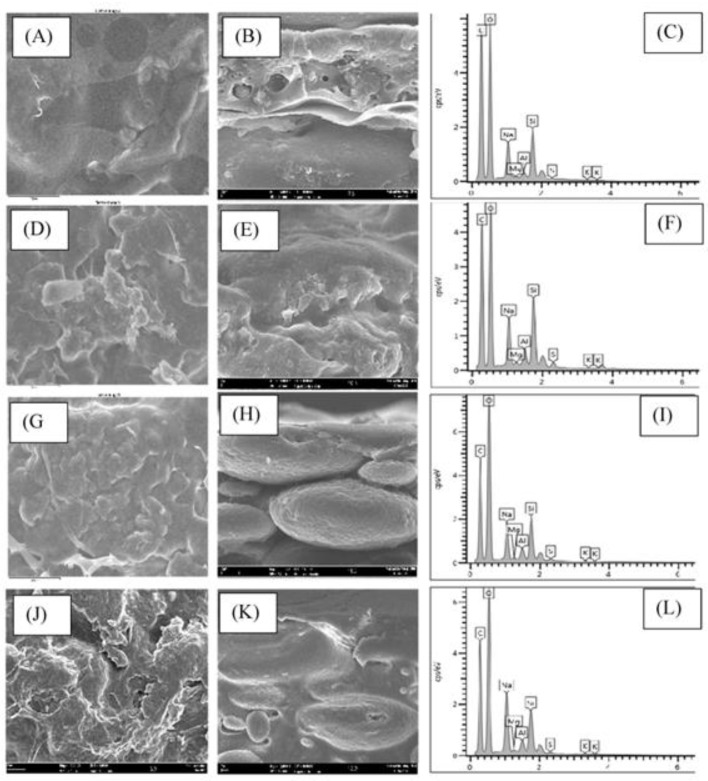
Surface image, cross-section, and graph of element quantity contained in the SPM membrane with variation of MMT loading at 0 wt% (**A**–**C**), 5 wt% (**D**–**F**), 10 wt% (**G**–**I**), and 20 wt% (**J**–**L**).

**Figure 5 polymers-15-02590-f005:**
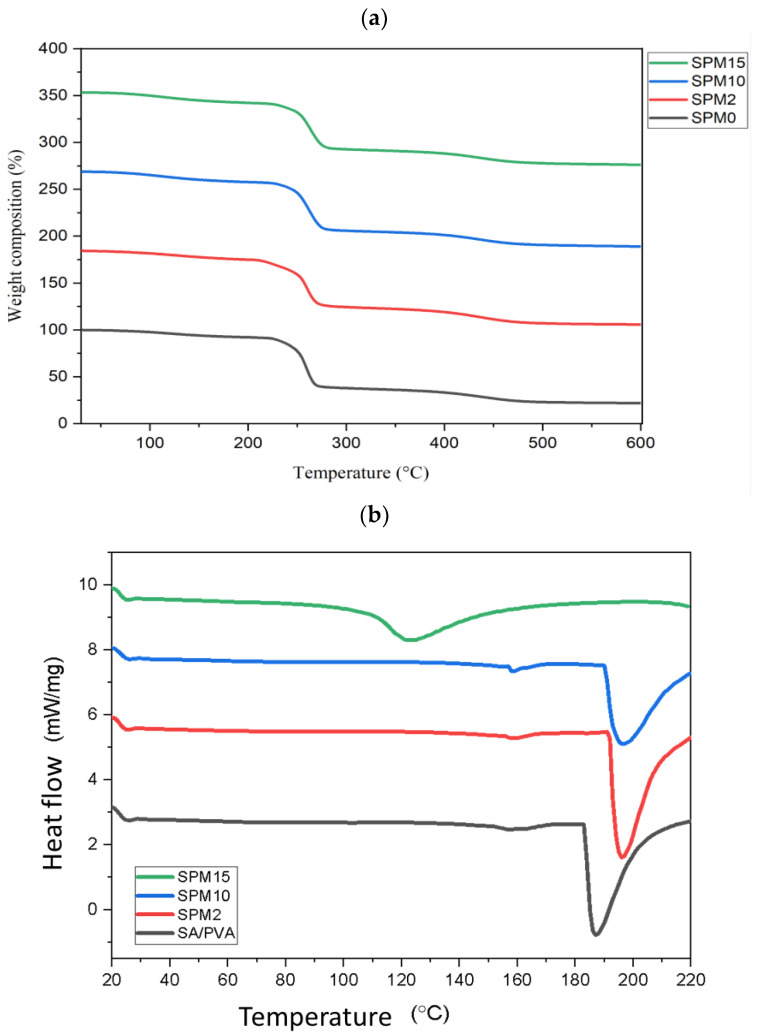
(**a**) TGA and (**b**) DSC analysis for SA/PVA-MMT membrane with various MMT loadings.

**Figure 6 polymers-15-02590-f006:**
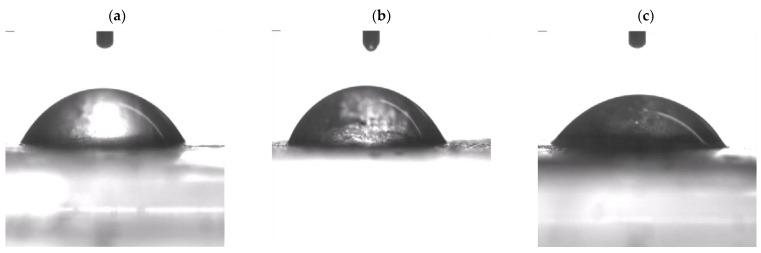
Contact angle analysis of various membranes (**a**) SA/PVA (control), (**b**) SPM5, (**c**) SPM10.

**Figure 7 polymers-15-02590-f007:**
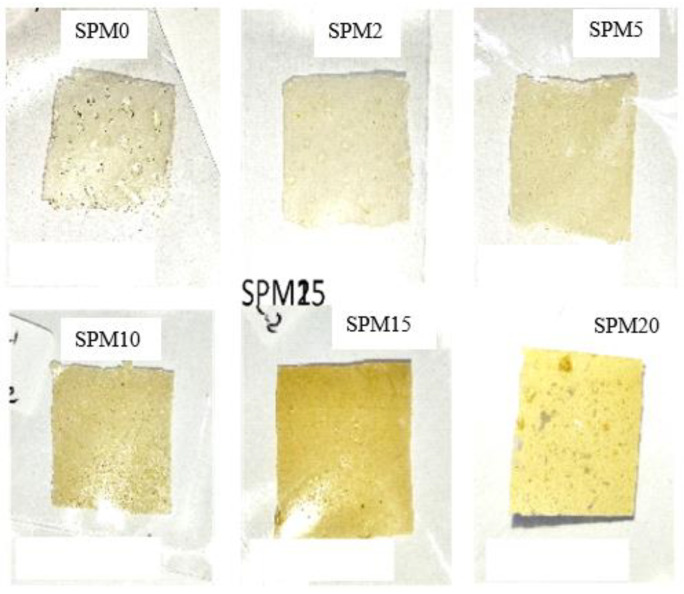
SVM membrane with MMT loading ranging from 0–20 wt%.

**Figure 8 polymers-15-02590-f008:**
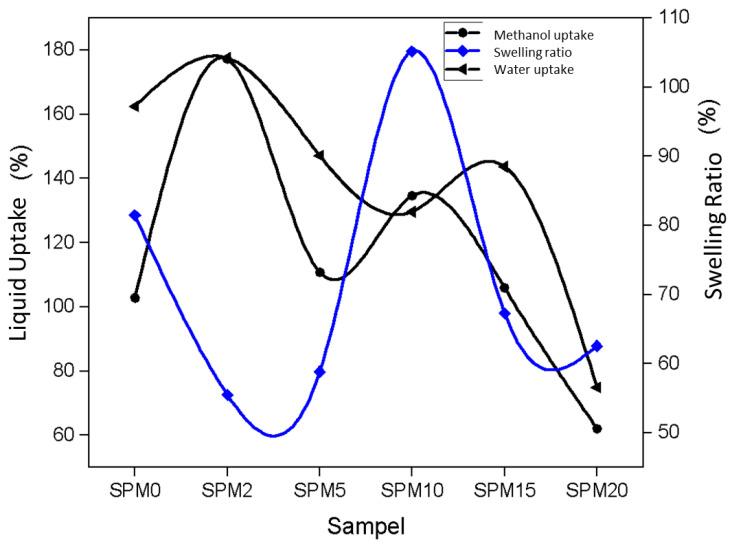
Liquid uptake and the swelling ratio at room temperature.

**Figure 9 polymers-15-02590-f009:**
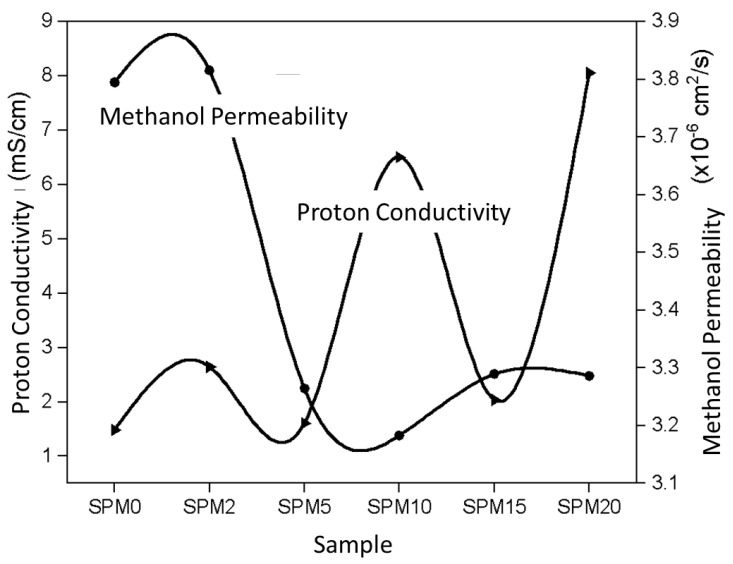
Proton conductivity and methanol permeability at room temperature.

**Figure 10 polymers-15-02590-f010:**
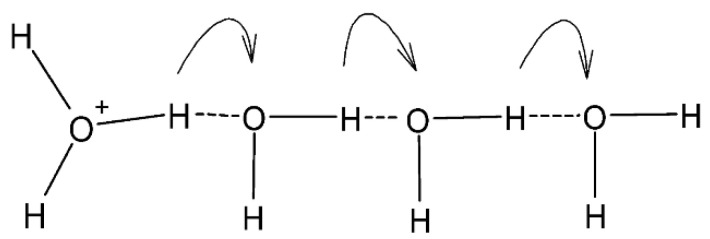
Grotthuss-type proton conductivity mechanism.

**Figure 11 polymers-15-02590-f011:**
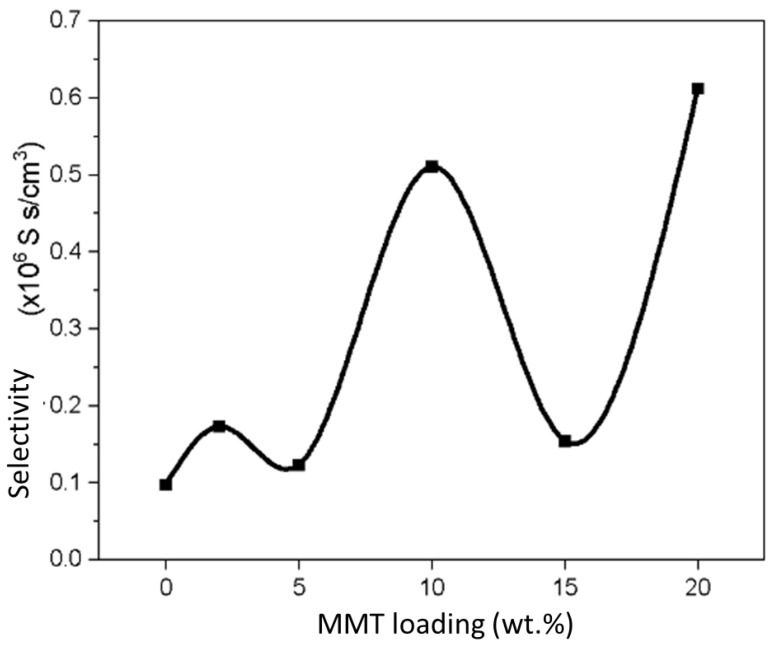
Selectivity of 0–20 wt% MMT in SA/PVA composites.

**Table 1 polymers-15-02590-t001:** Material content in various membranes.

Membrane	Ratio of SA:PVA	Content of MMT (wt%)
SPM0	40:60	0
SPM2	40:60	2
SPM5	40:60	5
SPM10	40:60	10
SPM15	40:60	15
SPM20	40:60	20

**Table 3 polymers-15-02590-t003:** Several performance results of the membrane with various loadings of MMT.

Sample	MMT Loading (wt%)	Water Uptake (%)	Methanol Uptake (%)	Swelling Ratio (%)
SPM0	0.0	162.46	98.55	81.45
SPM2	2.0	177.50	203.30	55.51
SPM5	5.0	147.10	132.23	58.84
SPM10	10.0	129.65	121.59	105.15
SPM15	15.0	143.70	97.46	67.27
SPM20	20.0	74.89	53.00	62.57

**Table 4 polymers-15-02590-t004:** Proton conductivity, methanol permeability, and selectivity for various membranes.

Sample	Proton Conductivity (mS/cm)	Methanol Permeability (×10^−8^ cm^2^/s)	Selectivity (×10^5^ S s/cm^3^)
SA/PVA	1.4828	1.5179	0.9769
SPM2	2.6429	1.5261	1.7318
SPM5	1.6083	1.3057	1.2318
SPM10	6.5025	1.2731	5.1077
SPM15	2.0332	1.3155	1.5455
SPM20	8.0510	1.3145	6.1250

## Data Availability

Available upon request.

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
