# Peer review of "Alginate/PVA Polymer Electrolyte Membrane Modified by Hydrophilic Montmorillonite for Structure and Selectivity Enhancement for DMFC Application"

_polymers, 2023, doi:10.3390/polym15122590_

Round 1

Reviewer 1 Report

Dear Authors

The current study is a continuation of efforts to find alternatives membrane for DMFC by producing sodium alginate/poly(vinyl alcohol) (SA/PVA) blended membrane with modification by montmorillonite (MMT) as an inorganic filler.

The following comments have to be considered by the authors:

General comments

The authors select one SA/PVA (40:60) composition based on what?

The authors did not declare why they used calcium chloride in the external crosslinking step?

The results of the proton conductivity and methanol permeability have fluctuated with variations in MMT concentration. The authors did not give a reasonable explanation. 

Specific comments

Abstract: The following sentence is not clear "due to the strong electrostatic attraction between H+, H3O+ and -OH ions, which H+ and the sodium alginate and PVA polymer matrix". Please revise and correct. 

2. Materials and methods 

2.1. Materials

The specifications of the used chemicals should be mentioned fully. 

2.2. Synthesis of SA/PVA biopolymer

The solvent used in the solvation of PVA and SA should be mentioned. Also, the volume of added GA should be mentioned. 

2.3. Synthesis of SA/PVA-MMT blended membrane

The sequence of added MMT to the SA/PVA/GA is not clear. The authors need to declare what have they added MMT, then dried as mentioned in section 2.2. Also, the volume of the poured SA/PVA-MMT and the size of the Petridish should be mentioned. 

3. Results and discussion

3.1. FTIR and Raman

The authors need to declare the effect of adding GA to SA/PVA solution.

Also, the combination of calcium chloride with GA at the second external crosslinking step should be investigated and confirmed. 

Recommendation

The authors need to major revision of the manuscript before reconsidering it for publication. 

The English of the manuscript in some parts needs polishing and in other parts needs rephrasing. 

Author Response

The reviewer's comments have been taken into action.

Reviewer 2 Report

1- why for the drying of the polymer mixture at 60 C, the furnace was used? for this temperature, an oven must be used. what is the reason to use a furnace?

2-why was the external crosslinking performed for the films? why the crosslinker agent was not added to the mixture initially?

3- how did the authors modify the thickness of the films? is it controllable for all the samples?

4-line 408: first it is said "All these membranes had a uniform thickness", then it is said," ....  thickened as the MMT increased". it is complicated. please clarify which sentence is correct.

5- Some references are too old. The authors can use the following references to improve the quality of discussion in the manuscript:

https://doi.org/10.1080/17425247.2022.2119220;

Sabbagh, F., Khatir, N. M., Karim, A. K., Omidvar, A., Nazari, Z., & Jaberi, R. (2019). Mechanical properties and swelling behavior of acrylamide hydrogels using montmorillonite and kaolinite as clays. J. Environ. Treat. Tech7(2), 211-219.

Author Response

(The authors gave the same response as above.)

Round 2

Reviewer 1 Report

Dear Authors

Thanks for your reply to the raised comments.

The revised version can be recommended for publication. 

A minor revision of the language is needed. 

Reviewer 2 Report

The comments are addressed properly, and the manuscript is acceptable in its current form.